# Compensation for Vanadium Oxide Temperature with Stereo Vision on Long-Wave Infrared Light Measurement

**DOI:** 10.3390/s22218302

**Published:** 2022-10-29

**Authors:** Chun-Yi Lin, Wu-Sung Yao

**Affiliations:** Department of Mechatronics Engineering, National Kaohsiung University of Science and Technology, Kaohsiung City 82445, Taiwan

**Keywords:** structured light stereo vision, binocular stereo vision, image interpolation, Planck’s blackbody radiation law, Stefan–Boltzmann law

## Abstract

In this paper, using automated optical inspection equipment and a thermal imager, the position and the temperature of the heat source or measured object can effectively be grasped. The high-resolution depth camera is with the stereo vision distance measurement and the low-resolution thermal imager is with the long-wave infrared measurement. Based on Planck’s black body radiation law and Stefan–Boltzmann law, the binocular stereo calibration of the two cameras was calculated. In order to improve the measured temperature error at different distances, equipped with Intel Real Sense Depth Camera D435, a compensator is proposed to ensure that the measured temperature of the heat source is correct and accurate. From the results, it can be clearly seen that the actual measured temperature at each distance is proportional to the temperature of the thermal image vanadium oxide, while the actual measured temperature is inversely proportional to the distance of the test object. By the proposed compensation function, the compensation temperature at varying vanadium oxide temperatures can be obtained. The errors between the average temperature at each distance and the constant temperature of the test object at 39 °C are all less than 0.1%.

## 1. Introduction

Accidents such as fires caused by old factory lines or machine failures are frequent, and occurrences in technology factories and electronics factories are more likely to affect the national or even global supply chain. The PCB industry is regarded as the most dangerous industry in the electronics industry because of its high-temperature process, complex pipelines, and many chemicals in the field. If the safety of the factory operation cannot be guaranteed, it will lead to a waste of human resources and high-value machines, which will lead to the failure of the factory to produce profits smoothly. The importance of industrial safety is the key to the stable growth of an enterprise. Visible or even invisible hidden dangers in the workshop must be detected and inspected to ensure the personal safety of operators and the property safety of high-value machines and equipment. Therefore, the demand for thermal imaging cameras for industrial use continues to increase. However, the cost of industrial thermal imaging cameras is much higher than that of other sensors. For most owners, it is a high investment, resulting in a low penetration rate in practical applications. Therefore, most factories still use manual labor. Regular testing results in personnel still having to travel to the front lines for operations, increasing the number of hours personnel are exposed to hazards and risks. In technical detection, the thermal imager mainly measures the surface temperature. Therefore, for many existing thermal imaging cameras in the market, temperature measurements with different depths of field cannot be achieved.

In this paper, automated optical inspection (AOI) equipment and a thermal imager heat source detection system are used. In order to improve the temperature measurement error at different distances, it is equipped with an Intel Real Sense Depth Camera D435, which can effectively grasp the distance from the heat source to the camera, and perform temperature distance compensation to ensure that the measured temperature is correct and accurate. In grasping the location and distance of the heat source, through the theoretical basis of Planck’s black body radiation law and Stefan–Boltzmann law, a deep camera with a thermal sense imaging system can be developed that can measure the temperature at different distances, so as to ensure that the temperature of the heat source can be captured.

According to the current thermal overflow phenomenon in chip packaging, the purpose is to more accurately compensate and improve the temperature measured at different distances through the extension results in this research. Additionally, through the method of non-single-point measurement of temperature, the far-infrared spectroscopy measurement of AOI in the future will be improved. Provide better compensation calculation methods for industrial accidents, packaging processes, wafer heating, and other projects.

This study is based on the calculation of the stereo vision distance measurement of the depth camera, the long-wave infrared measurement of the thermal imager, and the Planck black body radiation as the basis to obtain the exact temperature of the measured object at each distance. Through binocular stereo calibration, the corresponding positions of the two cameras are calculated to ensure that the measurement points are correct.

Optical stereo vision measurement technology can be divided into active measurement and passive measurement. Active measurement can be divided into continuous-wave lasers and pulsed lasers for time-of-flight ranging [1,2], triangular ranging and coded structured light, linear structured light, and spot structured light for structured light ranging [3]. Passive measurement includes the focusing method and defocusing method of monocular stereo vision, binocular stereo vision, and multi-eye stereo vision [4,5]. The difference between the active measurement and the passive measurement is that the active measurement uses infrared light or other signals that it emits and then receives back to obtain the actual distance to the object to be measured. The passive measurement uses the distance between the camera and the camera to calculate the actual distance to the object to be measured.

Binocular stereo calibration is based on the above-mentioned binocular stereo vision [6]. However, the lens pixels of the depth camera and the thermal imager are different. It is necessary to increase the image pixels of the thermal imager through the image interpolation method to achieve the same pixel condition of the two cameras. and there is a way to perform binocular stereo calibration. Image interpolation can be divided into nearest-neighbor interpolation, bilinear interpolation, and bicubic interpolation in linear interpolation. The edge information-based and wavelet coefficient-based can be found in nonlinear interpolation. For measurement requirements, the fastest nearest-neighbor interpolation method can be selected [7,8]. In this study, based on the data of the depth distance being known, the world coordinate system of the image captured by the camera is projected back to the camera coordinate system, the image coordinate system, and finally to the pixel coordinate system. Through the analysis of corresponding coordinate systems, the relationship between the depth camera and the thermal imager can be obtained.

The temperature measurement of the thermal imager in this study is based on the theory of blackbody radiation. Blackbody radiation is the electromagnetic radiation emitted by a blackbody in a thermodynamic equilibrium state, and the spectrum emitted by blackbody radiation depends on the temperature. It is well known that the energy wavelength of blackbody radiation is only related to temperature. The electromagnetic radiation emitted by a black body at the same temperature is higher than that of other objects. However, black bodies do not exist in real life, and the radiation in ideal and real life conditions is different, but the relationship between temperature and frequency corresponding to the black body radiation theory still exists. In this study, through the characteristics of Planck’s black body radiation law and Stefan–Boltzmann’s law [9,10], the degradation compensation coefficient of the measured object and the lens machine was obtained [11]. Take the black body radiant energy density whose wavelength is the independent variable as the measurement basis. By adding the wavelength-independent variable of the distance between the measured object and the lens, and adding the radiant energy density of the wavelength from vanadium oxide to germanium glass, the final measured temperature result can be obtained [12,13]. This study uses Planck’s law of blackbody radiation as the theoretical basis for thermal imaging, and extracts the far-infrared radiation information from the vanadium oxide sensing signal of each pixel in the thermal imager, and then converts it into temperature information. The attenuation of the far-infrared radiation wave during transmission is compensated by the Boltzmann constant and Planck’s black body radiation function [14,15].

The algorithm proposed in [16] can be effectively used for outlier detection and initial motion estimation in the RANSAC loop. The method can effectively solve the problem of image offset during depth measurements, and can further correct the offset problem of 3D images, which is a very critical image preprocessing method. The method of [17] can be used to solve more accurate 3D deformation, and to compensate for very complex environments such as underwater, polarized light, image distortion, light polarization, and other external influencing factors. This method can help this research to allow for the existence of more external imaging factors in future technological development, and ensure the conduct of the experiment.

In [18], a more accurate and suitable image localization method in various situations is proposed, and to achieve higher accuracy than traditional methods. The method [18] requires a small number of images for a short time to measure, which can be of great help for depth cameras. Especially, the frame rate of the depth camera reaches 60 fps, and through the technique of [18], the distance and the object under testing can be more effectively corrected repeatedly. The method of [18] can more effectively solve the problem when there is external environmental interference.

In a further in-depth discussion of the “object to be measured” purely based on the distance identification method, the method presented in [19] overcomes many challenges in complex lighting conditions, glare, light refraction, motion blur, and many other special problems. Ref. [19] provides Faster R-CNN, Cascade R-CNN, RetinaNet, YOLO-V3, CornerNet, and FCOS for image learning and comparison. It is an excellent solution for situations with a large number of images and a large amount of deformation of the object to be measured, as well as a large amount of glare or other factors that interfere with the measurement in the experimental environment. Since there are a large number of possible interference factors in the far-infrared radiation and the depth camera in this study when facing the object under test, the theory in this study will retain its possible assumptions, but the experimental environment must eliminate the above interference problems as much as possible.

In this study, the distance measurement experiment and distance compensation method will be carried out, and the distance of the camera will be verified at the same time by setting the temperature of the target as a constant in the experiment, measuring distances at different angles, and comparing the temperature differences before and after compensation. Among them are the original temperature of the contained vanadium oxide and the temperature of the target. The compensation method was carried out through the normal distribution of the error after the experiment. Through the compensation function obtained in this study, with more temperature data collection at different distances, the measured temperature at different distances can be effectively compensated, and a more accurate database analysis can be established. Through the proposed technology, it can be effectively used in an industrial safety inspection. Whether it is the capture of heat sources in indoor spaces or the control of product quality, the deep thermal imaging system proposed by this study can be used to overcome the temperature attenuation at different distances, and to measure the exact temperature of each point.

This research is mainly aimed at the limited hardware conditions of vanadium oxide and multi-vision depth cameras, and the proposed solution was also carried out with limited hardware resources. The main purpose was to conduct relevant research based on the equipment most used in the industry, rather than to carry out special experiments using special specifications or high-value equipment. Compared with the related literature studied in recent years, this paper can provide a contribution that can improve the regional temperature measurement in the stage of limited resources or periodic equipment replacement.

The outline of this paper is developed as follows. Section 2 describes the hardware of the selected depth camera, and performs transformation operations on the coordinate system corresponding to the image distance, and at the same time calculates the center point of the image to be measured, and then obtains the function of distance compensation. In Section 3, based on the hardware structure of the thermal imager, the temperature transfer function can be obtained according to Planck’s law of black body radiation and Stefan–Boltzmann’s law. In Section 4, experimental results are given to verify the effect of the proposed method. Finally in Section 5, concluding remarks are stated.

## 2. Image Processing Algorithm with Depth Camera

This study analyzed the depth camera setup, structured light projection, and dual-camera localization functions in the depth detection system. Using the Intel RealSense D435i depth camera, the distance between the measured object and the camera can be provided. By the conversion operation between the relevant coordinate systems of the binocular camera, the projected image of the overlap between the depth camera and the thermal imager can be obtained to locate the three-dimensional position information of the measured object.

The Intel RealSense D435i depth camera can be used to integrate RGB and depth images. However, to accurately obtain the depth and temperature information of the measured object, the stereo calibration and correction operations were further performed for the depth camera and thermal imager. Note that the horizontal and vertical FOVs of the two cameras are different. By confirming the camera parameters and corresponding positions through calibration, the distortion coefficient should be adjusted to ensure that the output images are not distorted.

### 2.1. Coordinate System Analysis of the Binocular Camera

Figure 1 is an integrated diagram of the four coordinate systems, i.e., pixel coordinate system, image coordinate system, camera coordinate system, and world coordinate system. The *O_W_*, *X_W_*, *Y_W_*, and *Z_W_* are represented as coordinate axes of the world coordinate system, mainly describing the location of the camera. The *O_C_*, *X_C_*, *Y_C_*, and *Z_C_* are denoted as coordinate axes of the camera coordinate system, and the origin *O_C_* is the optical center of the camera lens. The *o*, *x*, and *y* are the coordinate axes of the object image coordinate system, and the optical center of the camera lens is the midpoint of the object image. The *o_uv_*, *u*, and *v* are the coordinate axes of the pixel coordinate system, and the origin *o_uv_* is the upper left corner of the object image. The point *P* is given as any point in the world coordinate system. The point *p* is the imaging point, and its coordinates are represented by (*x*, *y*) in the image coordinate system and (*u*_0_, *v*_0_) in the pixel coordinate system. The *f* is the focal length of the camera, which is also equal to the distance between *O_C_* and *o*.

As shown in Figure 2, converting from the world coordinate system to the camera coordinate system, the object does not deform, so only translation and rotation are required. **R** is defined as the rotation matrix and **T** is defined as the translation vector. To convert the world coordinate system to the camera coordinate system, it is possible to apply rotation or translation, and rotate any angle around any coordinate axis to obtain the corresponding rotation matrix. Figure 3 shows a schematic diagram of ϕ degree rotation around the *z*-axis.

From Figure 3, we have
(1)$$\left[\begin{array}{l} x\\ y\\ z \end{array}\right]=\left[\begin{array}{ccc} \cos\phi & -\sin\phi & 0\\ \sin\phi & \cos\phi & 0\\ 0 & 0 & 1 \end{array}\right]\left[\begin{array}{l} x^{\prime }\\ y^{\prime }\\ z^{\prime } \end{array}\right]=R_{1}\left[\begin{array}{l} x^{\prime }\\ y^{\prime }\\ z^{\prime } \end{array}\right]$$

With ϑ degree rotation around the *x*-axis, we can obtain
(2)$$\left[\begin{array}{l} x\\ y\\ z \end{array}\right]=\left[\begin{array}{ccc} 1 & 0 & 0\\ 0 & \cos\vartheta & \sin\vartheta \\ 0 & -\sin\vartheta & \cos\vartheta \end{array}\right]\left[\begin{array}{l} x^{\prime }\\ y^{\prime }\\ z^{\prime } \end{array}\right]=R_{2}\left[\begin{array}{l} x^{\prime }\\ y^{\prime }\\ z^{\prime } \end{array}\right]$$

With ω degree rotation around the *y*-axis, we have
(3)$$\left[\begin{array}{l} x\\ y\\ z \end{array}\right]=\left[\begin{array}{ccc} \cos\omega & 0 & -\sin\omega \\ 0 & 1 & 0\\ \sin\omega & 0 & \cos\omega \end{array}\right]\left[\begin{array}{l} x^{\prime }\\ y^{\prime }\\ z^{\prime } \end{array}\right]=R_{3}\left[\begin{array}{l} x^{\prime }\\ y^{\prime }\\ z^{\prime } \end{array}\right]$$

Therefore, the rotation matrix **R** can be given as
(4)$$R=R_{1}R_{2}R_{3}$$

Additionally, the coordinate of the point *P* in the camera coordinate system is given as (5), where (5) can be rewritten as (6) with the translation vector **T**.
(5)$$\left[\begin{array}{l} X_{C}\\ Y_{C}\\ Z_{C} \end{array}\right]=R\left[\begin{array}{l} X_{W}\\ Y_{W}\\ Z_{W} \end{array}\right]+T$$
(6)$$\left[\begin{array}{c} X_{C}\\ Y_{C}\\ Z_{C}\\ 1 \end{array}\right]=\left[\begin{array}{cc} R & T\\ 0 & 1 \end{array}\right]\left[\begin{array}{c} X_{W}\\ Y_{W}\\ Z_{W}\\ 1 \end{array}\right]$$

For the camera coordinate system converted to the object image coordinate system, the three-dimensional space can be converted to two-dimensional space, as shown in Figure 4.

Through a similar triangle formula to (7), each coordinate can be calculated in equal proportion as shown in (8). The *x* and *y* of the point *p* in the image coordinate system can be obtained as (9). Then, the three-dimensional value of the camera coordinate system is calculated as (10).
(7)$$[\begin{array}{l} \Delta ABO_{C}\approx \Delta oCO_{C}\\ \Delta PBO_{C}\approx \Delta pCO_{C} \end{array}$$
(8)$$\frac{AB}{oC}=\frac{AO_{C}}{oO_{C}}=\frac{PB}{pC}=\frac{X_{C}}{x}=\frac{Z_{C}}{f}=\frac{Y_{C}}{y}$$
(9)$$x=f\frac{X_{C}}{Z_{C}},\;y=\frac{Y_{C}}{Z_{C}}$$
(10)$$Z_{C}\left[\begin{array}{l} x\\ y\\ 1 \end{array}\right]=\left[\begin{array}{cccc} f & 0 & 0 & 0\\ 0 & f & 0 & 0\\ 0 & 0 & 1 & 0 \end{array}\right]\left[\begin{array}{l} X_{C}\\ Y_{C}\\ Z_{C}\\ 1 \end{array}\right]$$

Both the object image coordinate system and the pixel coordinate system are defined on the same imaging plane; the only difference is the origin and unit of each coordinate system, as shown in Figure 5. The origin of the image coordinate system is located as the intersection of the imaging plane and the optical axis of the camera, and is also the midpoint of the imaging plane. Here, the *dx* and *dy* represent the millimeter length of each row and column, and one pixel is equal to *dx* millimeters.

Converting the image coordinate system (*x*, *y*) to the pixel coordinate system (*u*_0_, *v*_0_) can be given as (11). Equation (11) can be used to convert the image coordinate system to the pixel coordinate system. By (13), a point in the world coordinate system can be converted into the pixel coordinate system to obtain the corresponding pixel point.
(11)$$\left\{ \begin{array}{l} u=\frac{x}{dx}+u_{0}\\ v=\frac{y}{dy}+v_{0} \end{array}\right.$$
(12)$$\left[\begin{array}{l} u\\ v\\ 1 \end{array}\right]=\left[\begin{array}{ccc} \frac{1}{dx} & 0 & u_{0}\\ 0 & \frac{1}{dy} & v_{0}\\ 0 & 0 & 1 \end{array}\right]\left[\begin{array}{l} x\\ y\\ 1 \end{array}\right]$$
(13)$$Z_{C}\left[\begin{array}{l} u\\ v\\ 1 \end{array}\right]=\left[\begin{array}{cccc} f/dx & 0 & u_{0} & 0\\ 0 & f/dy & v_{0} & 0\\ 0 & 0 & 1 & 0 \end{array}\right]\left[\begin{array}{cc} R & T\\ \vec{0} & 1 \end{array}\right]\left[\begin{array}{l} X_{W}\\ Y_{W}\\ Z_{W}\\ 1 \end{array}\right]$$

### 2.2. Image Interpolation Method

With the obtained world coordinates of the depth camera, thermal imager, and the measured object, the shooting resolution of the two camera should be considered. To calculate the coordinate points corresponding to the pixel points, the resolution of the thermal imager must be increased to 1280 × 720, which is that of the depth camera, so image interpolation is required. The image interpolation method is used to enlarge the object image, which cannot be directly projected. Therefore, the values of these pixels need to be determined through the interpolation method, as shown in Figure 6.

The nearest neighbor interpolation method [20] is similar to the concept of rounding to an integer in one-dimensional space. In the two-dimensional space, the coordinates of the pixel points are represented by integers. This method mainly extracts the point closest to the target object, but a certain degree of loss of spatial symmetry is performed. In general, the pixel value of the closest integer coordinate point can be obtained by the point of the target image corresponding to the original image, and the final pixel value of the point can be obtained. As shown in Figure 7, the point *P* is the position corresponding to any point in the target object image to the original image. It can be seen that the closest point to point *P* is Q11, i.e., *f*(*P*) = *f*(*Q11*).

As can be seen from Figure 8, with the distances between the points being the same, the values of the surrounding pixels are the values of the original pixels. Although the nearest neighbor interpolation method has the fastest operation speed due to its simple conversion, it causes a mosaic phenomenon after zooming in on the image. However, this study is mainly applied to the single-point temperature measurement of the thermal imager; the conversion method of the nearest neighbor interpolation method can meet the requirements. The image pixels of the depth camera can be expanded from 160 × 120 to 1280 × 720 through image insertion, which is the same as the depth camera; the measured points corresponding to the two images can be grasped.

## 3. Long-wave Infrared Radiation Measurement and the Temperature Transfer Function

In this study, the thermal imaging vanadium oxide Lepton3 [13,14] produced by FLIR was selected as uncooled vanadium oxide. A plastic heating plate was used as the test object, and the main heating element was a brass thin plate. Considering that the thickness of the outer plastic plate is less than 0.5 mm, the emissivity of the plastic plate is difficult to be calculated [21]. The Stefan–Boltzmann law was used to calculate the net rate of radiative heat exchange between a black body and the surrounding medium [11]. However, for a surface that is not a black body, the radiation intensity of the spectrum does not completely follow the Planck distribution, and the radiation also radiates in a specific direction, so the Stefan–Boltzmann law of a non-black body can be given as
(14)$$Q_{radiation}=A\varepsilon \sigma \left(T_{S}{}^{4}-T_{A}{}^{4}\right)$$
where *A* is the surface area of the black body, *ε* is the emissivity of the radiating surface, *σ* is the Stefan-Boltzmann constant, *T_S_* is the absolute temperature of the black body, and *T_A_* is the absolute temperature of the surrounding medium. Emissivity is the ratio between surface emissivity and black body radiation at the same temperature. Material emissivity is generally between 0 and 1.0, with an ideal reflector having an emissivity of 0 and a black body emissivity of 1.0. Considering the material properties of emissivity, surface temperature, and smoothness, the emissivity of the test object in this study is 0.03. Note that the Stefan–Boltzmann constant is 5.67 × 10^−8^ Js^−1^ m^−2^ K^−4^.

The plastic heating plate of the test object in this study can be heated to a maximum temperature of 39 °C under a room temperature of 25 °C. Taking this into Wien’s displacement law [15], the wavelength peak value can be obtained as
(15)$$\lambda =\frac{2.89776829\times 10^{6}\;\mathrm{nm}\cdot K}{312\;K}=9288\;\mathrm{nm}$$

The sensitivity of the vanadium oxide used in this study is between 7000 and 14,000 nm, and when the Lepton radiation measurement mode is turned on, the 14-bit pixel value of the Lepton can remain stable. Since the radiation amount of the photosensitive band in vanadium oxide is linear, the image data provided by the thermal imager can also be linear. From the ambient temperature and the Planck curve, the output signal *S* of the thermal imager can be obtained as
(16)$$S=\int_{\lambda_{1}}^{\lambda_{2}}\frac{2\pi hc^{2}}{\lambda^{5}}\frac{1}{\exp\left(\frac{hc}{\lambda K_{B}T_{K}}\right)}R\left(\lambda \right)\cdot \delta \lambda$$
where *h* is Planck’s constant, *c* is the speed of light, *K_B_* is the Boltzmann constant, and *T_K_* is the absolute temperature. However, considering calculated in discrete data, (16) can be rewritten as
(17)$$T_{k}=\frac{B}{\ln\left(\frac{R}{S-O}+F\right)}$$
where *R*, *B*, *F*, and *O* are related parameters generated during calibration [22].
(18)$$[\begin{array}{l} R=\left[10000,1000000\right]\\ B=\left[1200,1700\right]\\ F=\left[0.5,3\right]\\ O=\left[-16384,16384\right] \end{array}$$

In this study, this thermal imager vanadium oxide system effectively measures the object at a distance of 0.4 to 0.8 m from the camera for calibration, and measures 1 to 3 m with high gain compensation. Taking the average value of the ROI in the image, the original expected range of the parameters of *R*, *B*, *F*, and *O* can be analyzed [23]. The value of the *F* is usually 1, unless the measured range is high than the ambient temperature, and the value of the *B* is 1428. The correction values for *R* and *O* are 231,160 and 6094.211 respectively. The target temperature *T_T_* (temperature in degrees Celsius) for each frame can be obtained as
(19)$$T_{T}=\frac{1428}{\ln\left(\frac{231160}{S-6094.211}+F\right)}+273.15$$

In this paper, the distance measurement experiment and distance compensation method was carried out, and the distance of the camera was verified at the same time, by setting the temperature of the target as a constant in the experiment, measuring distances at different angles, and comparing the temperature difference before and after compensation. Among them are the original temperature on the contained vanadium oxide and the temperature of the target. The compensation method was carried out through the normal distribution of the error after the experiment.

## 4. Experimental Analysis

Figure 9 shows the experimental setup. The FLIR development version of Lepton’s Uncooled Vox micro bolometer without the temperature calibration function was adopted, where the pixel size is 17 µm and the frames per second is 8.6 Hz. The thermal spectral range in longwave infrared was 8 µm to 14 µm. Temperature accuracy was in the temperature range of −10 °C to 140 °C. Thermal sensitivity was less than 50 mK. For shuttered configurations, the shutter assembly periodically blocks radiation from the scene and presents a uniform thermal signal to the sensor array, allowing an update to internal correction terms used to improve image quality. For applications in which there is little to no movement of the Lepton camera relative to the scene, the shutter assembly is recommended. The shutter assembly is less essential, although still capable of providing slight improvement to image quality, particularly at start-up or the ambient temperature varying rapidly. The shutter was also used as a reference for improved radiometric performance. Figure 10 shows the normalized response as a function of signal wavelength for the vanadium oxide array for each pixel.

In this study, a confined space was used and the vanadium oxide sensor was placed on the top as shown in Figure 11. The environment was not a vacuum but was an airtight environment, approaching the state of air molecules that are static and non-flowing. The surface of the circuit board was the chip mount and the chip to be tested, and the temperature sensor and other sensors were installed on the back. The bottom surface was a heating platform, and the infrared radiation of the heating platform was used for measurement. Because this enclosed space is not a vacuum during the measurement process, the infrared radiation can be absorbed by water molecules in the air. To reduce errors, the distance from the sensor to the platform should be smaller than 30 cm.

Through the proposed binocular vision integration, the overlapping images of the depth camera and the thermal imager were taken out to obtain the measurement center of the image. The image of Figure 12a was obtained by the visible camera, where the 445 datapoints were recorded. The image of Figure 12b was provided by the depth camera, where the distance of 1.5 m to the center point can be obtained. The black color area shows that the distance position cannot be located at the moment. The light blue part is the protrusion of the flat-panel heater protrusions. Figure 12c is an image of the thermal imager, where the numerical value of 30,141.5 is represented by $30,141.5\times 10^{-2}$ Kelvin degree.

The environmental conditions of the study were room temperature of 25 °C and humidity of 50%. The target object was a heating plate with a constant temperature of 39 °C. The reason why 39 degrees Celsius was selected as the constant test temperature is that the temperature change of 39 degrees Celsius is more constant and stable according to the change between the ambient temperature and the test panel. In addition, the vanadium oxide sensor used in this study has a low-gain data output between 25 degrees Celsius and 55 degrees Celsius, and therefore measurement accuracy will be higher, and it will be more helpful to the experiment. The main variables in this study are distance and vanadium oxide temperature. Therefore, the temperature of the tested body was selected that had the least influence on the overall experiment.

The temperature data at different distances can be obtained by adjusting the distance between the depth camera and the test object. We set five differential distances, i.e., 0.5, 1.0, 1.5, 2.0, 2.5 m, as the distance references. The collected data of the thermal imager were used to observe the influence of the vanadium oxide temperature on the actual temperature measurement. Note that 6000 pieces of data were obtained at each distance, and the actual temperature of the vanadium oxide was extracted for analysis.

Figure 13 is a comparison of the temperature data collected by the deep thermal imaging system at a distance of 0.5~2.5 m from the target object. The *x*-axis is the selected vanadium oxide temperature range of 25~40 °C, and the *y*-axis is the actual measured temperature. It can be found that the temperature is inversely proportional to the increase in the distance. Taking the target object at 39 °C, the compensation was performed for the temperature data difference at each distance, where the compensation functions of the distance at 0.5–2.5 m are given as (20). Through the obtained compensation Equation (20), the compensation results of the depth thermal imaging system at various distances can be shown in Figure 14.
(20)$$[\begin{array}{l} y_{0.5}=-5.595\ln(x)+19.14\\ y_{1.0}=-5.894\ln(x)+20.696\\ y_{1.5}=-5.343\ln(x)+19.281\\ y_{2.0}=-5.428\ln(x)+20.179\\ y_{2.5}=-6.427\ln(x)+24.084 \end{array}$$

As shown in Table 1, the temperatures of the vanadium oxide being 25–40 °C, the actual measured temperatures for varying vanadium oxide temperatures at a distance of 0.5 m are given. The average temperature is 39.986 °C, and the error is close to 0.035%.

As shown in Table 2, at the temperatures of the vanadium oxide being 25–40 °C, the actual measured temperatures for varying vanadium oxide temperatures at a distance of 1.0 m are given. The average temperature is 39.039 °C, and the error is about 0.1%.

As shown in Table 3, at the temperatures of the vanadium oxide being 25–40 °C, the actual measured temperatures for varying vanadium oxide temperatures at a distance of 1.5 m are given. The average temperature is 39.013 °C, and the error is about 0.033%.

As shown in Table 4, at the temperatures of the vanadium oxide being 25–40 °C, the actual temperature values measured for each vanadium oxide temperature at a distance of 2.0 m are averaged, and the temperature numerical errors are all between 39 ± 0.3, The total average temperature is 39.010 °C, and the error value is only 0.026%.

As shown in Table 5, the temperatures of the vanadium oxide being 25–40 °C, the actual measured temperatures for varying vanadium oxide temperatures at a distance of 2.5 m are given. The average temperature is 39.022 °C, and the error is only 0.056%.

From the results, it can be clearly seen that the actual measured temperature at each distance is proportional to the temperature of the thermal image vanadium oxide, while the actual measured temperature is inversely proportional to the distance of the test object. By the proposed compensation function at each distance, the compensation temperature at varying vanadium oxide temperatures can be obtained. As shown in Table 6, the errors between the average temperature at each distance and the constant temperature of the target object at 39 °C are all less than 0.1%.

## 5. Conclusions

This paper used structured light to project stereo vision in-depth vision to measure the distance from the measured (target) object. Through the center of the projection image of the depth camera and thermal imager with binocular stereo vision, the area required for measurement coordinates can be obtained. In the application of thermal imagers, based on Planck’s law of black body radiation and the Stefan–Boltzmann constant, it compensates for the attenuation of long-wave infrared transmission caused by the change of the distance.

In this study, under the environmental conditions of room temperature 25 °C and humidity 50%, with the temperature 39 °C of the target object, the thermal images of the vanadium oxide temperature 25–40 °C were analyzed. From the results, it can be clearly seen that the actual measured temperatures at each distance are proportional to the temperature of the thermal image of vanadium oxide, while the actual measured temperatures are inversely proportional to the distance of the target object. By the proposed compensation function, the compensation temperatures at varying vanadium oxide temperatures can be obtained. As shown in Table 6, the errors between the average temperature at each distance and the constant temperature of the target object at 39 °C are all less than 0.1%.

Through the compensation function obtained in this study, with more temperature data collection at different distances, the measured temperature at different distances can be effectively compensated, and a more accurate database analysis can be established. Through the proposed technology, it can be effectively used in an industrial safety inspection. Whether it is the capture of heat sources in indoor spaces or the control of product quality, the deep thermal imaging system proposed by this study can be used to overcome the temperature attenuation at different distances, and to measure the exact temperature of each point.

## Figures and Tables

**Figure 1 sensors-22-08302-f001:**
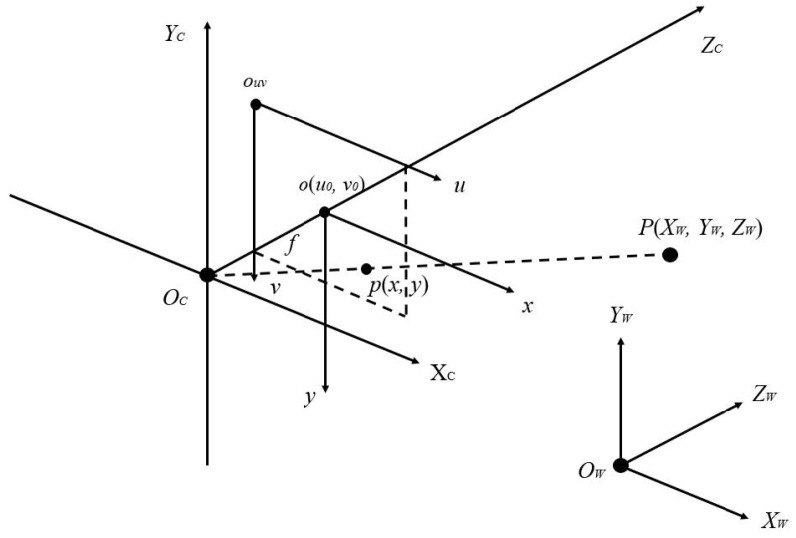
Integrated diagram of the four coordinate systems.

**Figure 2 sensors-22-08302-f002:**
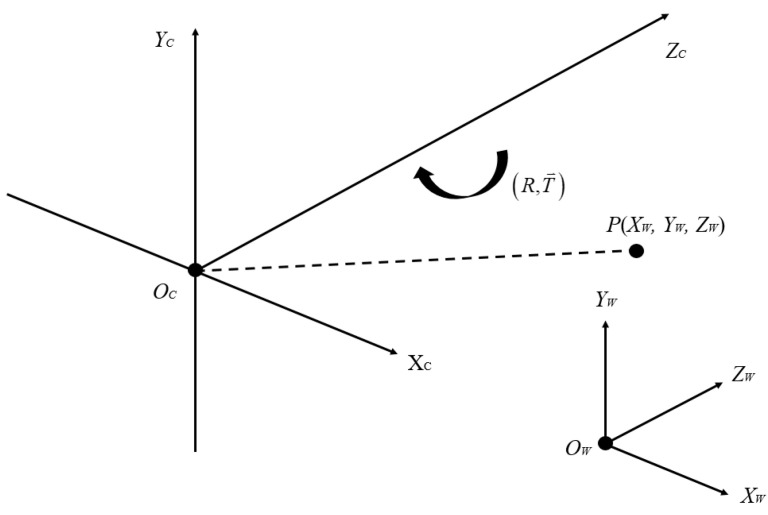
Relationship of world coordinate system and camera coordinate system.

**Figure 3 sensors-22-08302-f003:**
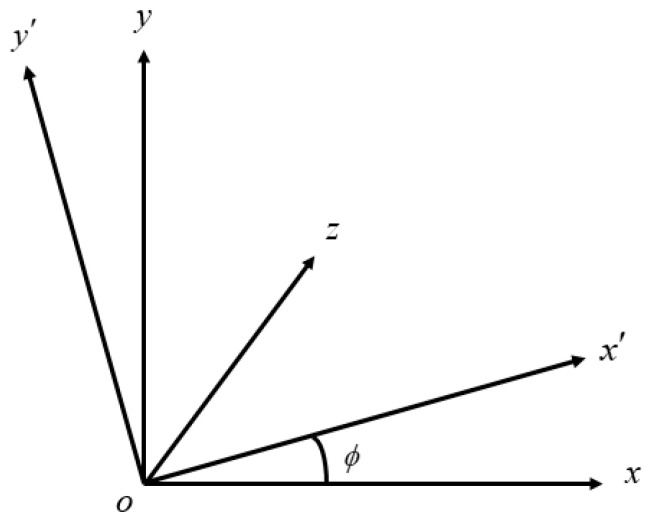
Schematic diagram of ϕ degree rotation around the *z*-axis.

**Figure 4 sensors-22-08302-f004:**
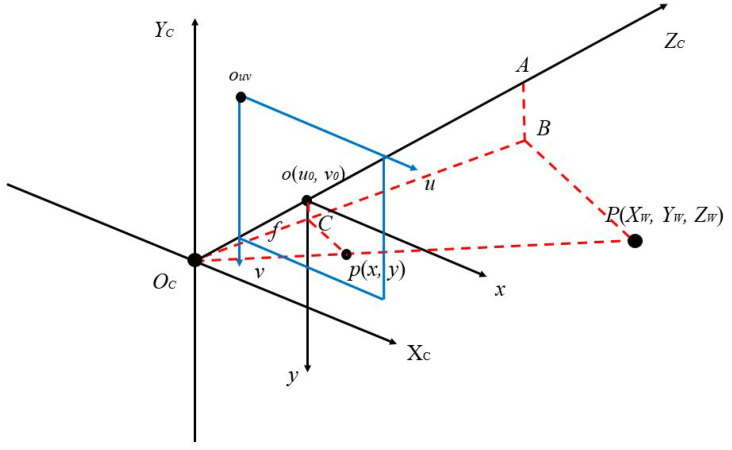
Relationship of camera coordinate system and image coordinate system.

**Figure 5 sensors-22-08302-f005:**
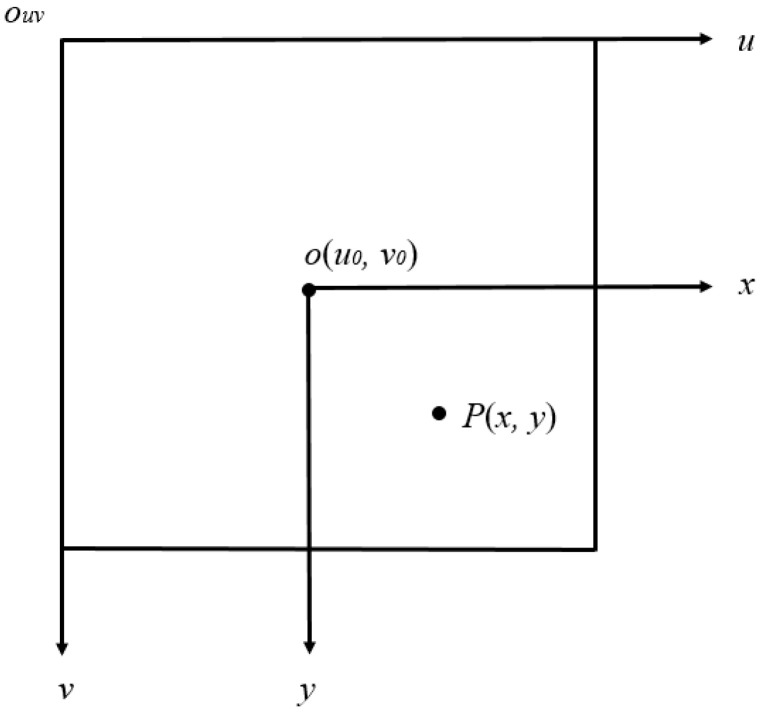
Relationship of image coordinate system and pixel coordinate system.

**Figure 6 sensors-22-08302-f006:**
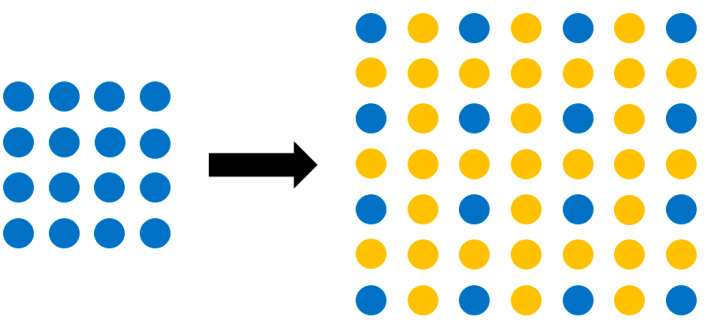
Schematic diagram of image interpolation method. The blue pixels are the original image, and the yellow pixels are the added pixels after processing.

**Figure 7 sensors-22-08302-f007:**
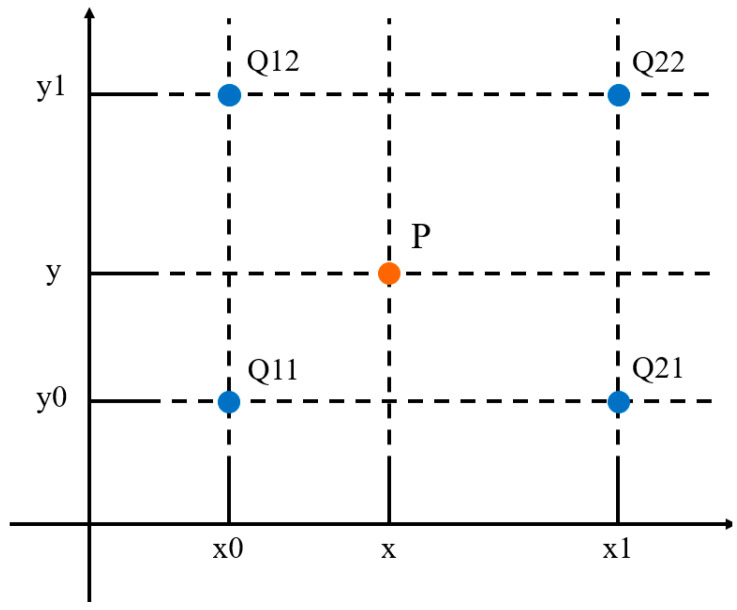
Characteristics of nearest neighbor interpolation method.

**Figure 8 sensors-22-08302-f008:**
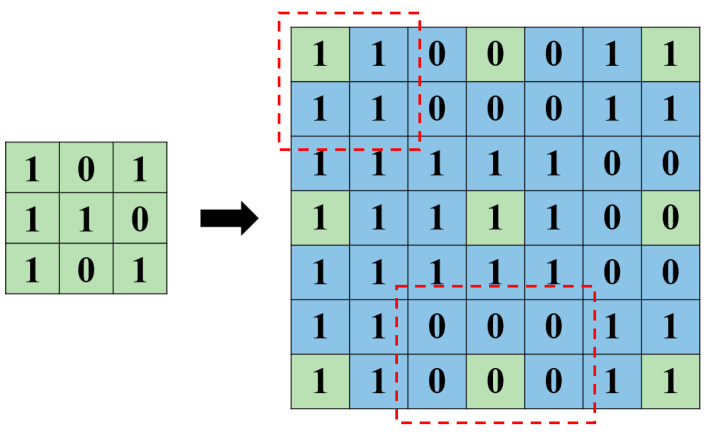
Schematic diagram of the nearest neighbor interpolation method with isometric interpolation.

**Figure 9 sensors-22-08302-f009:**
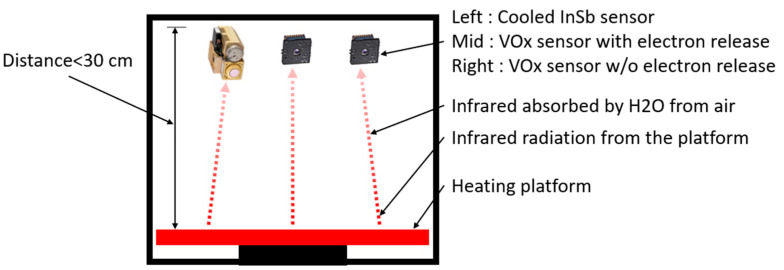
Experiment setup.

**Figure 10 sensors-22-08302-f010:**
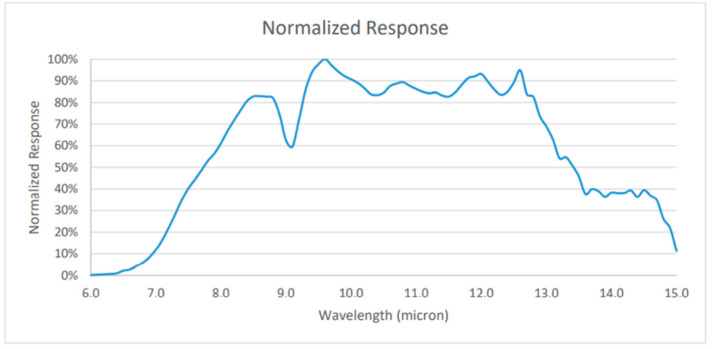
Normalized response of the VOx sensor.

**Figure 11 sensors-22-08302-f011:**
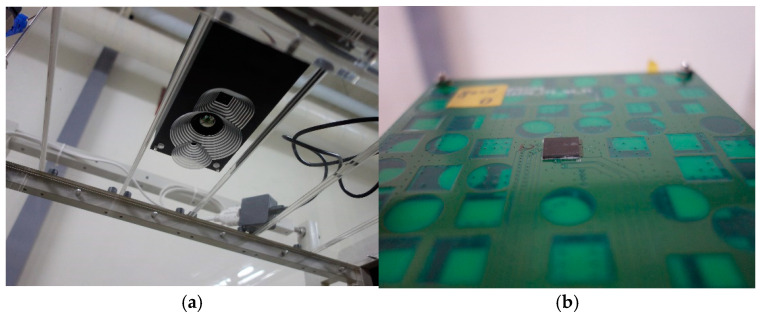
Experimental environment. (**a**) Experiment hardware including thermal radiation camera and visible light camera fixed in the wind tunnel. (**b**) Heating target platform and chip in the wind tunnel.

**Figure 12 sensors-22-08302-f012:**
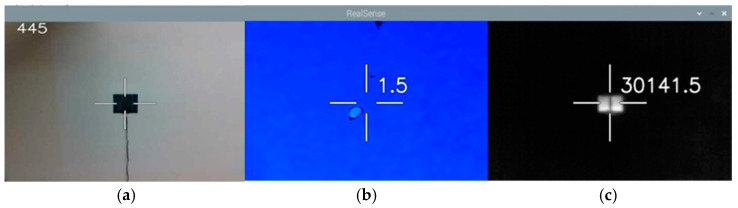
Imaging of the integrated thermal sensors and deep camera in the experiment. (**a**) Visible light camera image, (**b**) depth camera image, and (**c**) thermal imager image.

**Figure 13 sensors-22-08302-f013:**
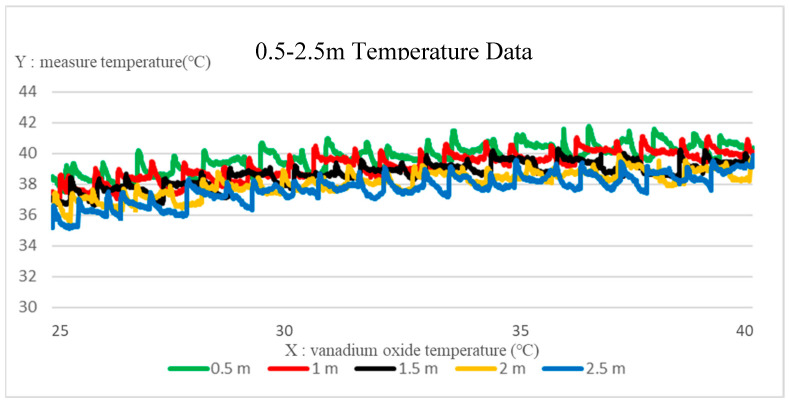
Comparison of the temperature data collected by the deep thermal imaging system without compensation.

**Figure 14 sensors-22-08302-f014:**
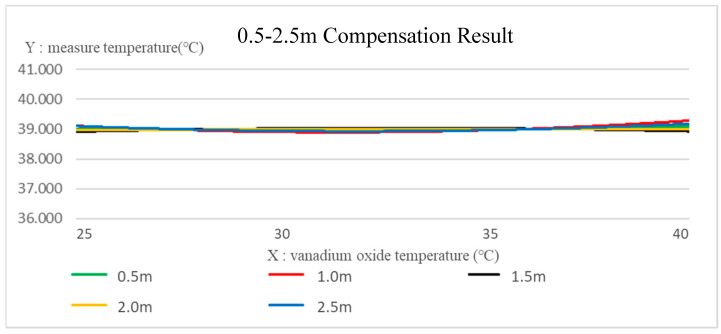
Compensation results of the depth thermal imaging system at various distances.

**Table 1 sensors-22-08302-t001:** Measured temperatures under varying vanadium oxide temperatures at 0.5 m.

**Vanadium Oxide Temperature**	25	26	27	28	29	30	31	32
**Measured Temperature**	39.302	39.308	38.943	38.716	38.982	38.791	38.733	39.183
**Vanadium Oxide Temperature**	33	34	35	36	37	38	39	40
**Measured Temperature**	38.983	38.726	39.142	38.821	39.073	39.171	39.076	38.831
**Average**	38.986							

**Table 2 sensors-22-08302-t002:** Measured temperatures under varying vanadium oxide temperatures at 1.0 m.

**Vanadium Oxide Temperature**	25	26	27	28	29	30	31	32
**Measured Temperature**	39.292	39.185	39.244	39.033	39.237	38.720	39.009	38.823
**Vanadium Oxide Temperature**	33	34	35	36	37	38	39	40
**Measured Temperature**	38.849	38.816	39.237	38.791	39.084	39.072	39.115	39.119
**Average**	39.039							

**Table 3 sensors-22-08302-t003:** Measured temperatures under varying vanadium oxide temperatures at 1.5 m.

**Vanadium Oxide Temperature**	25	26	27	28	29	30	31	32
**Measured Temperature**	38.889	39.187	38.905	39.180	39.040	38.925	39.003	38.769
**Vanadium Oxide Temperature**	33	34	35	36	37	38	39	40
**Measured Temperature**	39.301	38.797	38.985	38.971	39.244	39.232	38.735	39.040
**Average**	39.013							

**Table 4 sensors-22-08302-t004:** Measured temperatures under varying vanadium oxide temperatures at 2.0 m.

**Vanadium Oxide Temperature**	25	26	27	28	29	30	31	32
**Measured Temperature**	39.055	39.165	39.283	38.889	38.987	39.008	38.740	39.148
**Vanadium Oxide Temperature**	33	34	35	36	37	38	39	40
**Measured Temperature**	38.957	38.937	38.974	38.893	39.081	39.113	39.017	38.906
**Average**	39.010							

**Table 5 sensors-22-08302-t005:** Measured temperatures under varying vanadium oxide temperatures at 2.5 m.

**Vanadium Oxide Temperature**	25	26	27	28	29	30	31	32
**Measured Temperature**	39.024	38.996	39.097	39.150	39.124	38.834	39.026	39.152
**Vanadium Oxide Temperature**	33	34	35	36	37	38	39	40
**Measured Temperature**	38.835	39.061	39.113	38.777	38.934	39.094	38.897	39.232
**Average**	39.022							

**Table 6 sensors-22-08302-t006:** Measured temperatures and the errors under varying disturbances.

Distance (m)	Average Temperature (°C)	Error Percentage
0.5	38.986	0.035%
1.0	39.039	0.1%
1.5	39.013	0.033%
2.0	39.010	0.026%
2.5	39.022	0.056%

## Data Availability

Not applicable.

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
