# Peer review of "Compensation for Vanadium Oxide Temperature with Stereo Vision on Long-Wave Infrared Light Measurement"

_sensors, 2022, doi:10.3390/s22218302_

Round 1

Reviewer 1 Report

In this paper, using automated optical inspection equipment and a thermal imager, the  position and the temperature of the heat source or measured object can effectively be grasped. The high-resolution depth camera is with the stereo vision distance measurement and the low-resolution  thermal imager is with the long-wave infrared measurement. Based on Planck's black body radiation law and Stefan-Boltzmann law, the binocular stereo calibration of the two cameras is calculated.  In order to improve the measured temperature error at different distances, equipped with Intel Real Sense Depth Camera D435, a compensator is proposed to ensure that the measured temperature of the heat source is correct and accurate. From the results, it can be clearly seen that the actual measured temperature at each distance is proportional to the temperature of the thermal image vanadium oxide; while the actual measured temperature is inversely proportional to the distance of the  test object. By the proposed compensation function, the compensation temperature at varying vanadium oxide temperatures can be obtained. The error between the average temperature at each distance and the constant temperature of the test object at 39°C are all less than 0.1%. Generally, this is a quite interesting work. It can be accepted if the authors can consider the following issues:1. Why did the authors select 39°C as the test temperature? 2. Why did the authors select constant temperature? 3. What is the motivation of the work? Is there any other similar work in the literature? 4. More academic works are welcome to enrich the literature review such as Transnational image object detection datasets from nighttime driving, Model Reference Adaptive Control of Semi-active Suspension Model Based on AdaBoost Algorithm for Rollover Prediction. 5 More words are welcome for Figure 11. 6. The language should be improved as well.

Author Response

October 20, 2022

Paper Title:

Compensation for Vanadium Oxide Temperature with Stereo Vision on Long-Wave Infrared Light Measurement

Paper ID: sensors-1952784

Dear Editor

Thank you very much for your email dated October 11, 2022 regarding the Reviewers and Editor’s favorable comments on the above paper. I really appreciate the reviewer’s assistance to refine the manuscript. We have revised the paper in accordance with the comments. Detailed explanations are provided in the attached Response to Reviewers’ Comments. For your convenience, all of the corrections are marked in BLUE in the revised version.

Please let me know if there is any other question. I really appreciate your assistance during the reviewing process.

Yours sincerely,

Wu-Sung Yao, Ph.D.

Professor

Department of Mechatronics Engineering

National Kaohsiung University of Science and Technology

No 1 University Rd., Yanchao District, Kaohsiung 824, TAIWAN

wsyao@nkust.edu.tw

RESPONSE TO THE REVIEWER’S COMMENTS

‐‐‐‐‐‐‐‐‐‐‐‐‐‐‐‐‐‐‐‐‐‐‐‐‐‐‐‐‐‐‐‐‐‐‐‐‐‐‐‐‐‐‐‐‐‐

Reviewer 1

Comments: In this paper, using automated optical inspection equipment and a thermal imager, the  position and the temperature of the heat source or measured object can effectively be grasped. The high-resolution depth camera is with the stereo vision distance measurement and the low-resolution  thermal imager is with the long-wave infrared measurement. Based on Planck's black body radiation law and Stefan-Boltzmann law, the binocular stereo calibration of the two cameras is calculated.  In order to improve the measured temperature error at different distances, equipped with Intel Real Sense Depth Camera D435, a compensator is proposed to ensure that the measured temperature of the heat source is correct and accurate. From the results, it can be clearly seen that the actual measured temperature at each distance is proportional to the temperature of the thermal image vanadium oxide; while the actual measured temperature is inversely proportional to the distance of the  test object. By the proposed compensation function, the compensation temperature at varying vanadium oxide temperatures can be obtained. The error between the average temperature at each distance and the constant temperature of the test object at 39°C are all less than 0.1%. Generally, this is a quite interesting work. It can be accepted if the authors can consider the following issues:1. Why did the authors select 39°C as the test temperature? 2. Why did the authors select constant temperature? 3. What is the motivation of the work? Is there any other similar work in the literature? 4. More academic works are welcome to enrich the literature review such as Transnational image object detection datasets from nighttime driving, Model Reference Adaptive Control of Semi-active Suspension Model Based on AdaBoost Algorithm for Rollover Prediction. 5 More words are welcome for Figure 11. 6. The language should be improved as well.

Response:

  1. In this paper, the distance measurement experiment and distance compensation method will be carried out, and the distance of the camera will be verified at the same time. By setting the temperature of the target as a constant in the experiment. Measure distances at different angles. And compare the temperature difference before and after compensation. Among them are the original temperature on the contained vanadium oxide and the temperature of the target. The compensation method is carried out through the normal distribution of the error after the experiment. More explanations have been given in the revised version. Please refer to lines 363 to 369 on page 12 of the revised version.

  1. The reason to determine 39 degrees Celsius as the constant test temperature is that the temperature change of constant 39 degrees Celsius is more stable according to the change between the ambient temperature and the test panel. In addition, the vanadium oxide sensor used in this study has a low-gain data output between 25 degrees Celsius and 55 degrees Celsius, and the measurement accuracy can be higher, and it is more helpful to the experiment. The main variables in this study are distance and vanadium oxide temperature. Therefore, the temperature of the tested body is selected in the experiment with the constant that has the least influence on the overall experiment. Please refer to lines 363 to 369 on page 12.

  1. According to the current thermal overflow phenomenon in chip packaging, the purpose is to more accurately compensate and improve the temperature measured at different distances through the extension results in this research. And through the method of non-single-point measurement of temperature, the far-infrared spectroscopy measurement of AOI in the future will be improved. Provide better compensation calculation methods for industrial accidents, packaging processes, wafer heating, and other projects. More statements can be referred to lines 54 to 59 on page 2 of the revised version.

  1. Four references have been included in the revised version, as shown below. More related illustrations have been given. Please refer to pages 3 and page 4 of the revised version.

[16] Banglei Guan, Ji Zhao, Zhang Li, Fang Sun, and Friedrich Fraundorfer, “Relative Pose Estimation with a Single Affine Correspondence”, IEEE Transactions on Cybernetics, Vol. 52, No.10, pp. 10111-10122, 2022.

[17] Zhilong Su, Jiyu Pan, Shui qiang, Zhang, Shen Wu, Qi feng Yu, and Dong sheng Zhang, “Characterizing Dynamic Deformation of Marine Propeller Blades with Stroboscopic Stereo Digital Image Correlation”, Mechanical Systems and Signal Processing, Vol. 162, 108072, 2022.

[18] Ziyi Jin, Zhixue Li, Tianyuan Gan, Zuoming Fu, Chongan Zhang, Zhongyu He, Hong Zhang, Peng Wang, Jiquan Liu, and Xuesong Ye, “A Novel Central Camera Calibration Method Recording Point-to-Point Distortion for Vision-Based Human Activity Recognition”, Sensors 2022, Vol.22, No. 9, 3524, 2022.

[19] Chang Nie, Muhammad Ali Qadar, Shaodong Zhou, Hui Zhang, Yang Shi, Jinwu Gao, and Zhifeng Sun, “Transnational Image Object Detection Datasets from Nighttime Driving”, Signal, Image and Video Processing, 2022.

  1. In this revised version, more detailed content and descriptions are added. Please refer to the description of Fig. 11 on page 11.

  1. The English writing of the revised version has been improved.

Reviewer 2 Report

This MS reports measuring the temperature of the vanadium oxide by combining stereo vision and long-wave infrared imaging. Though the technique is helpful in some advanced industrial fabrications, the contribution of the work is not clear due to some significant issues. So I cannot recommend publishing the MS on Sensors unless the following issues could be addressed:

1. In the Introduction, the background information presented by the authors (with only 19 reference citations) lags far behind the current progress. It is clear that for such a topic involving stereo vision and infrared imaging only 1 citation in the past two years cannot provide a clear acknowledgment of the related work and, further, cannot statement the potential impact of the work. As a suggestion, the authors should consult a comprehensive literature review of the related work, not least that for those work in the past two years. Here are some recommendations:

[1] Guan et al, Relative pose estimation with a single affine correspondence, 2021, IEEE TOC.

[2] Su et al,  Characterizing dynamic deformation of marine propeller blades with stroboscopic stereo digital image correlation, 2022, MSSP.

[3] Jin et al,  A Novel Central Camera Calibration Method Recording Point-to-Point Distortion for Vision-Based Human Activity Recognition, 2022, Sensors.

2. In Section 2, coordinate system transformation and image interpolation are well-established basics in stereo vision and optical measurement and can be found in many textbooks, such as the well-known Multiple View Geometry in Computer Vision and Digital Image Processing. So it is not necessary to repeat them in detail in the MS.

3. Similarly, Section 3 simply reused the previous work, I cannot find any innovation or improvement. Please introduce the original contribution of the MS.

4. In the part of experimental analysis, though the results are presented in detail, the figures must be improved to meet the requirement of a journal paper but not an experiment report.

Author Response

October 20, 2022

Paper Title:

Compensation for Vanadium Oxide Temperature with Stereo Vision on Long-Wave Infrared Light Measurement

Paper ID: sensors-1952784

Dear Editor

Thank you very much for your email dated October 11, 2022 regarding the Reviewers and Editor’s favorable comments on the above paper. I really appreciate the reviewer’s assistance to refine the manuscript. We have revised the paper in accordance with the comments. Detailed explanations are provided in the attached Response to Reviewers’ Comments. For your convenience, all of the corrections are marked in BLUE in the revised version.

Please let me know if there is any other question. I really appreciate your assistance during the reviewing process.

Yours sincerely,

Wu-Sung Yao, Ph.D.

Professor

Department of Mechatronics Engineering

National Kaohsiung University of Science and Technology

No 1 University Rd., Yanchao District, Kaohsiung 824, TAIWAN

wsyao@nkust.edu.tw

Reviewer 2

Comments: This MS reports measuring the temperature of the vanadium oxide by combining stereo vision and long-wave infrared imaging. Though the technique is helpful in some advanced industrial fabrications, the contribution of the work is not clear due to some significant issues. So I cannot recommend publishing the MS on Sensors unless the following issues could be addressed:

  1. In the Introduction, the background information presented by the authors (with only 19 reference citations) lags far behind the current progress. It is clear that for such a topic involving stereo vision and infrared imaging only 1 citation in the past two years cannot provide a clear acknowledgment of the related work and, further, cannot statement the potential impact of the work. As a suggestion, the authors should consult a comprehensive literature review of the related work, not least that for those work in the past two years. Here are some recommendations:

[1] Guan et al, Relative pose estimation with a single affine correspondence, 2021, IEEE TOC.

[2] Su et al, Characterizing dynamic deformation of marine propeller blades with stroboscopic stereo digital image correlation, 2022, MSSP.

[3] Jin et al, A Novel Central Camera Calibration Method Recording Point-to-Point Distortion for Vision-Based Human Activity Recognition, 2022, Sensors.

  1. In Section 2, coordinate system transformation and image interpolation are well-established basics in stereo vision and optical measurement and can be found in many textbooks, such as the well-known Multiple View Geometry in Computer Vision and Digital Image Processing. So it is not necessary to repeat them in detail in the MS.

  1. Similarly, Section 3 simply reused the previous work, I cannot find any innovation or improvement. Please introduce the original contribution of the MS.

  1. In the part of experimental analysis, though the results are presented in detail, the figures must be improved to meet the requirement of a journal paper but not an experiment report.

  • Response:
  1. Four references as shown below have been included in the revised version. More related illustrations and discussions are rewritten. Please refer to pages 3 and page 4 of the revised version.

In [16], it presented four cases of minimal solutions for two-view relative pose estimation by exploiting the affine transformation between feature points, and it demonstrates efficient solvers for these cases. It is shown that under the planar motion assumption or with knowledge of a vertical direction, a single affine correspondence is sufficient to recover the relative camera pose. The four cases considered are two-view planar relative motion for calibrated cameras as a closed-form and least-squares solution, a closed-form solution for an unknown focal length, and the case of a known vertical direction. These algorithms can be used efficiently for outlier detection within a RANSAC loop and for initial motion estimation. All the methods are evaluated on both synthetic data and real-world datasets. The experimental results demonstrate that our methods outperform comparable state-of-the-art methods in accuracy with the benefit of a reduced number of needed RANSAC iterations. According to [16], the problem of localization drift in multi-vision cameras can be further strengthened.

The stereo digital image correlation (stereo-DIC) technique proposed in [17] shows a powerful capacity for full-field three-dimensional (3D) deformation measurement of rotary machine structures. However, it remains a challenge for underwater applications due to difficulties in imaging, stereo calibration, light refraction, and so on. In this paper, its potential in characterizing 3D dynamic deformation of the underwater rotor blades is shown by solving these problems. A stroboscopic stereo-DIC system is established to capture clear speckle images of a blade that rotates in the windowed cavitation tunnel under different flow speeds. To calibrate the stereo camera for underwater object measurement, an improved planar pattern-based calibration method and a globally optimal relative pose estimation algorithm are proposed to calibrate the intrinsic and extrinsic parameters of the stereo-DIC system separately. In particular, a novel refractive 3D reconstruction method for underwater objects is presented to recover the true 3D shape according to flat refraction geometry, ensuring the correctness and reliability of the measured 3D displacement fields. Several experiments demonstrated that the proposed stereo-DIC system and methods are feasible and accurate. Based on this, the measured dynamic displacement fields of a propeller blade are presented and discussed. Results herald a possibility for monitoring the full-field 3D dynamic response and structural health of underwater rotating structures by the proposed technique.

In [18], a more accurate and more suitable method for image positioning is proposed in a variety of situations. The camera is the main sensor of vision-based human activity recognition, and its high precision calibration of distortion is an important pre-requisite of the task. Current studies have shown that multi-parameter model methods achieve higher accuracy than traditional methods in the process of camera calibration. However, these methods need hundreds or even thousands of images to optimize the camera model, which limits their practical use. Here, we propose a novel point-to-point camera distortion calibration method that requires only dozens of images to get a dense distortion rectification map. It have designed an objective function based on deformation between the original images and the projection of reference images, which can eliminate the effect of distortion when optimizing camera parameters. Dense features between the original images and the projection of the reference images are calculated by digital image correlation (DIC). Experiments indicate that our method obtains a comparable result with the multi-parameter model method using a large number of pictures, and contributes a 28.5% improvement to the reprojection error over the polynomial distortion model. However, although this method in [18] presents the most accurate and efficient way, it still has to limit the hardware in the experiment to meet the original situational assumptions of this study.

In a further in-depth discussion of the "object to be measured" purely based on the distance identification method, the literature [19] discussed the Vision-based autonomous driving systems need to overcome many challenges at nighttime, such as complicated illumination conditions, dazzle caused by headlamps, light refraction, motion blur, and many other special problems. Although many research works have gradually paid attention to low-light challenges, there is still lacking natural nighttime driving datasets covering various countries and regions. Thus, we propose the transnational image object detection datasets from nighttime driving (TDND datasets), which contain natural driving images across multiple countries and regions. The TDND datasets not only cover severe weather such as heavy rain and snow, but also retain complicated illumination conditions and other problems. These datasets consist of 115k images which are annotated in six classes. The performance of six deep-learning based object detection methods is further compared for evaluation, which are Faster R-CNN, Cascade R-CNN, RetinaNet, YOLO-V3, CornerNet, and FCOS. The results show that the quality of the TDND datasets is comparable to that of MS-COCO. Moreover, for special problems at nighttime, the state-of-the-art object detection methods are worthy of further research and optimization. It proposes the transnational image object detection datasets from nighttime driving (TDND datasets). These datasets collect driving images across multiple countries and regions from natural scenes, which include many visual special problems of nighttime driving, such as complicated illumination conditions, dazzle caused by headlamps, light refraction, and motion blur. Moreover, these datasets cover deep nighttime, dusk, daytime and include various weather like rain, snow, and sunny. In addition, the TDND datasets are statistically analyzed, which are separated by different regions for better hyperparameters setting and model training. To evaluate the proposed datasets, six typical object detection methods are trained on them. The analysis shows that the performance of typical object detectors is poor at the nighttime driving scene. This means that most of the current object detection methods are principally designed for daytime. So the proposed nighttime driving datasets can help the development of object detection methods at nighttime scenes. It is known that nighttime driving occupies a large part of the driving time. However, there are still many special challenges at the nighttime driving scene that need to be explored and resolved. Therefore, we believe that the TDND datasets can further promote frontier research in the field of nighttime vision.

In this study, the distance measurement experiment and distance compensation method will be carried out, and the distance of the camera will be verified at the same time. By setting the temperature of the target as a constant in the experiment. Measure distances at different angles. And compare the temperature difference before and after compensation. Among them are the original temperature on the contained vanadium oxide and the temperature of the target. The compensation method is carried out through the normal distribution of the error after the experiment. Through the compensation function obtained in this study, with more temperature data collection at different distances, the measured temperature at different distances can be effectively compensated, and a more accurate database analysis can be established. By the proposed technology, it can be effectively used in an industrial safety inspection. Whether it is the capture of heat sources in indoor spaces or the control of product quality, the deep thermal imaging system proposed by this study can be used to overcome the temperature attenuation at different distances, and to measure the exact temperature of each point.

This research is mainly aimed at the limited hardware conditions of vanadium ox-ide and multi-vision depth cameras, and the proposed solution is also carried out with limited hardware resources. The main purpose is to conduct relevant research based on the equipment most used in the industry, rather than to carry out special experiments using special specifications or high value equipment. Compared with the related literatures studied in recent years, this paper can provide the contribution that can improve the regional temperature measurement in the stage of limited resources or periodic equipment replacement.

[16] Banglei Guan, Ji Zhao, Zhang Li, Fang Sun, and Friedrich Fraundorfer, “Relative Pose Estimation with a Single Affine Correspondence”, IEEE Transactions on Cybernetics, Vol. 52, No.10, pp. 10111-10122, 2022.

[17] Zhilong Su, Jiyu Pan, Shui qiang, Zhang, Shen Wu, Qi feng Yu, and Dong sheng Zhang, “Characterizing Dynamic Deformation of Marine Propeller Blades with Stroboscopic Stereo Digital Image Correlation”, Mechanical Systems and Signal Processing, Vol. 162, 108072, 2022.

[18] Ziyi Jin, Zhixue Li, Tianyuan Gan, Zuoming Fu, Chongan Zhang, Zhongyu He, Hong Zhang, Peng Wang, Jiquan Liu, and Xuesong Ye, “A Novel Central Camera Calibration Method Recording Point-to-Point Distortion for Vision-Based Human Activity Recognition”, Sensors 2022, Vol.22, No. 9, 3524, 2022.

[19] Chang Nie, Muhammad Ali Qadar, Shaodong Zhou, Hui Zhang, Yang Shi, Jinwu Gao, and Zhifeng Sun, “Transnational Image Object Detection Datasets from Nighttime Driving”, Signal, Image and Video Processing, 2022.

  1. Thanks to the reviewer's suggestion, the duplicate article paragraphs have been removed. Because the relevant information is cited in the experiment, some important information is still retained in the text. Please refer to page 5 line 213.

This research is mainly aimed at the limited hardware conditions of vanadium oxide and multi-vision depth cameras, and the proposed solution is also carried out with limited hardware resources. The main purpose is to conduct relevant research based on the equipment most used in the industry, rather than to carry out special experiments using special specifications or high value equipment. Compared with the related literature studied in recent years, this paper can provide the contribution that can improve the regional temperature measurement in the stage of limited resources or periodic equipment replacement.

  1. Thanks to the reviewer's suggestion, the duplicate article paragraphs have been removed. Because the relevant information will be cited in the experiment in the text, some important information is still retained in the text. Please refer to page 12 line 363.

In this paper, the distance measurement experiment and distance compensation method will be carried out, and the distance of the camera will be verified. By setting the temperature of the target as a constant in the experiment. The distances can be measured at different angles. And compare the temperature difference before and after compensation. Among them are the original temperature on the contained vanadium oxide and the temperature of the target. The compensation method is carried out through the normal distribution of the error after the experiment.

  1. In this revised version, more detailed content and descriptions are added. Please refer to the description of pages 17 and 18. Through the compensation function obtained in this study, with more temperature data collection at different distances, the measured temperature at different distances can be effectively compensated, and a more accurate database analysis can be established. Through the proposed technology, it can be effectively used in an industrial safety inspection. Whether it is the capture of heat sources in indoor spaces or the control of product quality, the deep thermal imaging system proposed by this study can be used to overcome the temperature attenuation at different distances, and to measure the exact temperature of each point.

Reviewer 3 Report

The problems in the paper are as follows:

1.The references are short of the literature in recent three years, so it is recommended to add relevant contents.

2.Missing in the introduction of a subsection or paragraph in which the contribution is explained compared to other works.

3.In the introduction, comments on various works, but the authors do not mention how these works differ from their own. 

4.What is the general title of Figure 12?

Author Response

October 20, 2022

Paper Title:

Compensation for Vanadium Oxide Temperature with Stereo Vision on Long-Wave Infrared Light Measurement

Paper ID: sensors-1952784

Dear Editor

Thank you very much for your email dated October 11, 2022 regarding the Reviewers and Editor’s favorable comments on the above paper. I really appreciate the reviewer’s assistance to refine the manuscript. We have revised the paper in accordance with the comments. Detailed explanations are provided in the attached Response to Reviewers’ Comments. For your convenience, all of the corrections are marked in BLUE in the revised version.

Please let me know if there is any other question. I really appreciate your assistance during the reviewing process.

Yours sincerely,

Wu-Sung Yao, Ph.D.

Professor

Department of Mechatronics Engineering

National Kaohsiung University of Science and Technology

No 1 University Rd., Yanchao District, Kaohsiung 824, TAIWAN

wsyao@nkust.edu.tw

Reviewer 3

Comments: The problems in the paper are as follows:

1.The references are short of the literature in recent three years, so it is recommended to add relevant contents.

2.Missing in the introduction of a subsection or paragraph in which the contribution is explained compared to other works.

3.In the introduction, comments on various works, but the authors do not mention how these works differ from their own.

4.What is the general title of Figure 12?

  • Response:
  1. Many thanks to the reviewer’s comments. Four references as shown below have been included in the revised version. More related illustrations and discussions are rewritten. Please refer to pages 3 and page 4 of the revised version.

[16] Banglei Guan, Ji Zhao, Zhang Li, Fang Sun, and Friedrich Fraundorfer, “Relative Pose Estimation with a Single Affine Correspondence”, IEEE Transactions on Cybernetics, Vol. 52, No.10, pp. 10111-10122, 2022.

[17] Zhilong Su, Jiyu Pan, Shui qiang, Zhang, Shen Wu, Qi feng Yu, and Dong sheng Zhang, “Characterizing Dynamic Deformation of Marine Propeller Blades with Stroboscopic Stereo Digital Image Correlation”, Mechanical Systems and Signal Processing, Vol. 162, 108072, 2022.

[18] Ziyi Jin, Zhixue Li, Tianyuan Gan, Zuoming Fu, Chongan Zhang, Zhongyu He, Hong Zhang, Peng Wang, Jiquan Liu, and Xuesong Ye, “A Novel Central Camera Calibration Method Recording Point-to-Point Distortion for Vision-Based Human Activity Recognition”, Sensors 2022, Vol.22, No. 9, 3524, 2022.

[19] Chang Nie, Muhammad Ali Qadar, Shaodong Zhou, Hui Zhang, Yang Shi, Jinwu Gao, and Zhifeng Sun, “Transnational Image Object Detection Datasets from Nighttime Driving”, Signal, Image and Video Processing, 2022.

  1. Thanks to the reviewer's suggestion, the duplicate article paragraphs have been added. Please refer to line 118 to 198 on page 4. This research is mainly aimed at the limited hardware conditions of vanadium oxide and multi-vision depth cameras, and the proposed solution is also carried out with limited hardware resources. The main purpose is to conduct relevant research based on the equipment most used in the industry, rather than to carry out special experiments using special specifications or high value equipment. Compared with the related literature studied in recent years, this paper can provide the contribution that can improve the regional temperature measurement in the stage of limited resources or periodic equipment replacement.

  1. More detailed content and descriptions are added, please refer to the line 199 to 212 on page 4. In this study, the distance measurement experiment and distance compensation method will be carried out, and the distance of the camera will be verified at the same time. By setting the temperature of the target as a constant in the experiment. Measure distances at different angles. And compare the temperature difference before and after compensation. Among them are the original temperature on the contained vanadium oxide and the temperature of the target. The compensation method is carried out through the normal distribution of the error after the experiment. Through the compensation function obtained in this study, with more temperature data collection at different distances, the measured temperature at different distances can be effectively compensated, and a more accurate database analysis can be established. Through the proposed technology, it can be effectively used in an industrial safety inspection. Whether it is the capture of heat sources in indoor spaces or the control of product quality, the deep thermal imaging system proposed by this study can be used to over-come the temperature attenuate at different distances, and to measure the exact temperature of each point.

  1. Thanks to the reviewer for the correction, the general title has been added in the revised version. Please refer to line 14 in page 415. Imaging of the integrated thermal sensors and deep camera in the experiment (a) Visible light camera image, (b) depth camera image, and (c) thermal imager image.

Round 2

Reviewer 2 Report

Please don't copy from the original papers directly for the newly added content in the introduction. 

Author Response

Reviewer 2

Comments: Please don't copy from the original papers directly for the newly added content in the introduction.

  • Response:
  1. Thanks to the reviewer for the comments. Modifications have been done as shown below. Please refer to pages 3 lines 110-137 of the revised version.

The algorithm proposed in [16] can be effectively used for outlier detection and initial motion estimation in the RANSAC loop. The method can effectively solve the problem of image offset during depth measurement, and can further correct the offset problem of 3D images, which is a very critical image preprocessing method. The method of [17] can be used to solve more accurate 3D deformation, and to compensate for very complex environments such as underwater, polarized light, image distortion, light polarization, and other external influencing factors. This method can help this research to allow the existence of more external imaging factors in future technological development, and ensure the conduct of the experiment.

In [18], a more accurate and suitable image localization method in various situations is proposed, and to achieve higher accuracy than traditional methods. The method [18] requires a small number of images for a short time to measure, which can be of great help for depth cameras. Especially, the frame rate of the depth camera reaches 60 fps, and through the technique of [18], the distance and the object under test can be more effectively corrected repeatedly. The method of [18] can more effectively solve the problem when there is external environmental interference.

In a further in-depth discussion of the "object to be measured" purely based on the distance identification method. The method presented in [19] overcomes many challenges in complex lighting conditions, glare, light refraction, motion blur, and many other special problems. [19] provides Faster R-CNN, Cascade R-CNN, RetinaNet, YOLO-V3, CornerNet, and FCOS for image learning and comparison. It is an excellent solution for situations with a large number of images and a large amount of deformation of the object to be measured, as well as a large amount of glare or other factors that interfere with the measurement in the experimental environment. Since there are a large number of possible interference factors in the far-infrared radiation and the depth camera in this study when facing the object under test, the theory in this study will retain its possible assumptions, but the experimental environment must eliminate the above interference problems as much as possible.

Round 3

Reviewer 2 Report

Accept in present form.